# Severe Complications after General Anesthesia versus Sedation during Pediatric Diagnostic Cardiac Catheterization for Ventricular Septal Defect

**DOI:** 10.3390/jcm11175165

**Published:** 2022-08-31

**Authors:** Yuki Ogawa, Hayato Yamana, Tatsuya Noda, Miwa Kishimoto, Shingo Yoshihara, Koshiro Kanaoka, Hiroki Matsui, Kiyohide Fushimi, Hideo Yasunaga, Masahiko Kawaguchi, Tomoaki Imamura

**Affiliations:** 1Department of Public Health, Health Management and Policy, Nara Medical University, Nara 634-8521, Japan; 2Department of Anesthesiology, Nara Medical University, Nara 634-8521, Japan; 3Department of Health Services Research, Graduate School of Medicine, The University of Tokyo, Tokyo 113-0033, Japan; 4Japan Medical Office, Takeda Pharmaceutical Company Limited, Tokyo 103-8668, Japan; 5Regional Healthcare Planning Division, Health Policy Bureau, Ministry of Health, Labour and Welfare, Tokyo 100-8916, Japan; 6Department of Medical and Health Information Management, National Cerebral and Cardiovascular Center, Osaka 564-8565, Japan; 7Department of Clinical Epidemiology and Health Economics, School of Public Health, The University of Tokyo, Tokyo 113-0033, Japan; 8Department of Health Policy and Informatics, Tokyo Medical and Dental University Graduate School, Tokyo 113-8510, Japan

**Keywords:** cardiac catheterization, anesthesia, ventricular septal defect

## Abstract

Pediatric cardiac catheterization requires unconsciousness and immobilization through general anesthesia or sedation. This study aimed to compare the occurrence of severe complications in pediatric diagnostic cardiac catheterization for ventricular septal defect between general anesthesia and sedation performed under similar institutional environments. Using the Japanese Diagnosis Procedure Combination database, we retrospectively identified pediatric patients (aged <2 years) who underwent diagnostic cardiac catheterization for ventricular septal defect between July 2010 and March 2019. The composite outcome was the occurrence of severe complications, including catecholamine use and intensive care unit admission, within seven days after catheterization. Overlap weighting based on propensity scores was used to adjust for patient- and hospital-level confounding factors. We identified 3159 patients from 87 hospitals, including 930 under general anesthesia and 2229 under sedation. The patient- and hospital-level baseline characteristics differed between the groups. After adjustment, the proportion of patients with severe complications was significantly higher in the general anesthesia group than in the sedation group (2.4% vs. 0.6%; risk difference, 1.8% [95% confidence interval, 0.93–2.6%]). Severe complications occurred more frequently in the general anesthesia group than in the sedation group. Further research on anesthetic methods is necessary to assess the safety and accuracy of pediatric diagnostic cardiac catheterization.

## 1. Introduction

Cardiac catheterization is often performed in children with congenital heart disease. In Japan, children with ventricular septal defect (VSD) receive catheterization for preoperative evaluation. Complications caused by a vascular puncture, catheter manipulation, instrument malfunction, and anesthesia are reported to occur in 4–10% of pediatric catheterization cases [1,2,3,4,5]. Severe complications, such as death, emergency surgery, extracorporeal membrane oxygenation (ECMO), intensive care unit (ICU) admission, and unplanned tracheal intubation, occur in less than 1% of cases [2,4].

Children should be unconscious and immobilized during cardiac catheterization by either general anesthesia with airway management or sedation with intravenous anesthesia [3]. It is recommended that procedural sedation be performed within an institutional environment (i.e., patient monitoring and emergency support), consistent with the requirements for general anesthesia [6]. Several studies have compared the incidence of complications between general anesthesia and sedation. A single-center cohort study reported a lower incidence of complications within the sedated group compared to the group receiving general anesthesia [7,8]. An observational study from eight institutions also compared the occurrence of severe complications after pediatric cardiac catheterization between general anesthesia (6%) and sedation (2%) [9].

However, the relationship between anesthetic methods and the occurrence of severe complications after pediatric cardiac catheterization requires further assessment. Large-scale studies involving multiple institutions are necessary to evaluate the generalizability of the findings of previous studies. In addition, according to the Catheterization Risk Score for Pediatrics, there are several risk factors for severe complications: emergency catheterization, age, weight, administration of circulatory agonists, respiratory status, organ dysfunction, American Society of Anesthesiologists classification, and procedure [4]. To evaluate the effects of anesthetic methods on the occurrence of severe complications, studies need to compare the general anesthesia and sedation performed within the same procedure under similar institutional environments with adjusted patient backgrounds.

This study aimed to compare the occurrence of severe complications in pediatric cardiac catheterization between general anesthesia and sedation performed under similar institutional environments. We analyzed patients with stable conditions undergoing diagnostic catheterization using a nationwide database in Japan.

## 2. Materials and Methods

### 2.1. Ethical Considerations

This study was approved by the institutional review board of the University of Tokyo. The requirement for written informed consent from patients was waived by the review board because of the retrospective design of this study and the anonymous nature of the data. This manuscript adheres to the applicable STROBE guidelines.

### 2.2. Data Source

This retrospective study used the Diagnosis Procedure Combination database, a nationwide database of discharge abstracts and claims data for acute care inpatients in Japan. The database includes approximately 7 million inpatients annually from approximately 1100 participating hospitals, which covers 50% of all acute care inpatients in Japan [10]. The database includes the following data: hospital identification number; patient age, sex, body weight, and height; main diagnosis; comorbidities at admission; and complications after admission recorded using the International Classification of Diseases, 10th Revision (ICD-10) (codes and text data in Japanese); surgical and nonsurgical procedures recorded with the original Japanese coding system; procedure dates; dates of drug administration; length of stay; and discharge status. Physicians are responsible for accurately recording patient data at discharge with reference to medical records because these records are linked to the payment and reimbursement system [11]. We also obtained the hospital characteristics from the 2014 Annual Report for the Functions of Medical Institutions.

### 2.3. Patients

We retrospectively identified pediatric patients (aged < 2 years) who underwent diagnostic cardiac catheterization for VSD (ICD-10 code: Q21.0) between July 2010 and March 2019. We excluded the following patients: emergency admissions, hospitalization for >7 days prior to catheterization, ICU stays before the day of catheterization, mechanical ventilation before the day of catheterization, VSD closure within 7 days of catheterization, and missing data on body weight. It is common for pediatric patients with stable conditions to be discharged after undergoing diagnostic catheterization for VSD. Patients who underwent VSD closure immediately after catheterization were likely to be unstable before catheterization. Therefore, this group of patients was excluded. We merged patient-level data with hospital characteristic data from the Annual Report for Functions of Medical Institutions. Patients admitted to the following hospitals were excluded: hospitals from which the data could not be merged with the Annual Report for Functions of Medical Institutions and hospitals without an ICU. The anesthetic method used during catheterization (general anesthesia or sedation) was determined according to the Japanese code for procedures. Claims for sedation can be reimbursed when sedation with loss of consciousness is performed in an institutional environment that is sufficient for general anesthesia if necessary. Patients with no claims data for anesthesia were excluded from the analysis.

### 2.4. Outcomes

The composite outcome was the occurrence of severe complications within 7 days of catheterization. Following previous studies [4,7] and the established definitions for complications [2], we defined severe complications as any of the following: death, surgery for cardiac tamponade, pericardiocentesis, cardiopulmonary resuscitation, electrical cardioversion, ECMO, catecholamine (adrenaline, noradrenaline, dopamine, or dobutamine) use, or ICU admission. We excluded ICU admission within 1 day of catheterization from the outcome because patients may be routinely monitored in the ICU after catheterization.

### 2.5. Statistical Analysis

To adjust for the measured confounding factors between the general anesthesia and sedation groups, we used overlap weighting based on propensity scores [12,13]. First, we estimated the propensity scores using a logistic regression model with the anesthetic method (general anesthesia or sedation) as the dependent variable. The following patient-level variables were used for estimating the propensity scores: age, sex, body weight, comorbidities at admission (pulmonary hypertension [ICD-10: I27.0, I27.2, and P29.3], heart failure [ICD-10: I11.0, I50.0, I50.1, I50.9, and P29.0], and 21-, 18-, or 13-trisomy [ICD-10: Q90, Q91.0-3, and Q91.4-7]), and year of catheterization. We also used the following hospital characteristics reported in the 2014 Annual Report for Functions of Medical Institutions: annual hospital volume of all surgeries, number of beds, number of ICU beds, number of pediatric beds, number of pediatric-related ICU beds, category of pediatric inpatient management fee based on the number of pediatricians and nurses and hospital systems, and hospital category (1, academic hospital; 2, nonacademic advanced hospital; or 3, other hospitals). Hospital volume and number of beds were categorized according to the hospital-level medians. The C-statistics were calculated to evaluate the discriminatory ability of the model. Each patient was then weighted by the predicted probability of receiving the opposite treatment. We used standardized mean differences to assess the balance in the characteristics after weighting. An absolute value of a standardized mean difference exceeding 10% was considered to indicate an imbalance in the patient background [14].

The composite outcomes of severe complications were compared between the general anesthesia and sedation groups. Risk differences and their 95% confidence intervals (CI) were calculated using a weighted, generalized linear model with binomial distribution and an identity link function.

We used a significance level of *p* < 0.05 for all statistical tests. All statistical analyses were performed using Stata/SE 17.0 (Stata Corp., College Station, TX, USA).

## 3. Results

We identified 6379 pediatric patients who underwent diagnostic cardiac catheterization for VSD during the study period. After excluding 1305 patients based on patient factors, 677 patients based on hospital factors, and 1238 patients without claims data for anesthesia, 3159 eligible patients from 87 hospitals were included. There were 930 and 2229 patients in the general anesthesia and sedation groups, respectively (Figure 1). No patient was recorded with both codes of general anesthesia and sedation.

Table 1 shows the baseline characteristics of all eligible patients (*n* = 3159). Before adjustment, patients in the general anesthesia group were older and more likely to have trisomy (21, 18, or 13) than those in the sedation group. Compared to the sedation group, the general anesthesia group had more patients admitted to hospitals with more ICU beds and fewer patients admitted to advanced hospitals.

After adjustment, the baseline characteristics were well-balanced between the groups (Table 2). The C-statistic was 0.77.

The outcomes in both the groups before and after adjustment are compared in Table 3. Before adjustment, the proportion of patients with severe complications was significantly higher in the general anesthesia group than in the sedation group (2.6% vs. 0.8%, *p* = 0.001; risk difference, 1.8% [95% CI, 0.74–2.9%]). After adjustment, the proportion of severe complications was also significantly higher in the general anesthesia group than in the sedation group (2.4% vs. 0.6%, *p* < 0.001; risk difference, 1.8% [95% CI, 0.93–2.6%]).

The types of severe complications before and after the adjustment are shown in Table 4. The most severe complications were catecholamine use and ICU admission. Death, cardiac tamponade surgery, and ECMO management did not occur in the study patients. The median duration of catecholamine use was 3 (interquartile range [IQR], 1.75–4) and 4 (IQR, 2–7) days in the general anesthesia and sedation groups, respectively.

## 4. Discussion

This study investigated the association between anesthetic methods and severe complications in pediatric patients (aged < 2 years) who underwent diagnostic cardiac catheterization for VSD. We used a nationwide database in Japan and conducted a propensity score analysis to adjust for the patient and hospital characteristics. Our results showed that the occurrence of severe complications was significantly higher with general anesthesia than with sedation when performed in comparable institutional environments.

Pediatric patients undergoing cardiac catheterization for congenital heart disease are at a high risk of complications due to cardiac and non-cardiac complications. In previous studies, general anesthesia was selected for patients with risk factors, including younger age, lower weight, complex cardiac malformations, and genetic syndromes. In addition, severe complications were associated with the type of procedure (diagnostic, biopsy, and interventional) [2,15]. Therefore, a study adjusting for patients’ backgrounds and the type of procedure is required. VSD is a common indication for pediatric cardiac catheterization, and its severity and treatment can be generally distinguished by the patient’s age. Cardiac catheterization for VSD in patients younger than 2 years is mostly performed for preoperative examination and not for treatment. Therefore, we analyzed pediatric patients younger than 2 years who underwent diagnostic cardiac catheterization for VSD to reduce selection bias.

We conducted a large-scale study using a nationwide database and adjusted for hospital-level confounding factors and patient characteristics. Hospital-level variables are important because a previous study showed that the choice of sedation or general anesthesia for pediatric catheterizations was institution-dependent [9]. Furthermore, the risk averseness of the facilities was associated with a lower incidence of complications [16]. We defined the composite outcome by following previous studies [4,7] and the established definitions [2]. Although death, cardiac tamponade surgery, and ECMO management did not occur in the study patients, electrical cardioversion, catecholamine use, and ICU admission represented the severe complications of the VSD patients who were in stable conditions before the procedure. In our study, after adjustment, general anesthesia was associated with an increased proportion of severe complications as compared to sedation. This result was similar to those of previous studies, despite the differences in patient criteria and adjustment for confounding factors.

We used overlap weighting [12,13] based on propensity scores to adjust for confounding factors. Overlap weighting is a propensity score method that attempts to mimic randomized clinical trials, enabling covariate balance and statistical power in a clinically relevant target population. Overlap weighting assigns weights to each patient in proportion to the probability that a patient belongs to the opposite group and emphasizes the comparison of patients at clinical equipoise.

Of the 4397 patients with data, 1238 (28%) had no claims data for anesthesia. This implies that some catheterizations were performed under minimal sedation or without sufficient preparation to administer general anesthesia if an emergency occurred. In this study, we compared general anesthesia and sedation after excluding these patients because pediatric catheterization requires unconsciousness and a sufficient level of patient monitoring and emergency support [3,6]. Further research is necessary to evaluate the safety of catheterization performed under potentially inappropriate sedation.

A possible mechanism for the increased severity of complications in the general anesthesia group is that the most common complication in pediatric cardiac catheterization is hypotension, followed by upper airway obstruction, hypoxia, and hypercapnia [9]. General anesthesia requires more anesthetic drugs than sedation and therefore poses a higher risk of hypotension during the procedure. Hypotension may lead to circulatory collapse [17]. Although general anesthesia with airway management causes fewer respiratory complications (airway obstruction and hypoventilation) than sedation, the effect of hypotension may be stronger.

The results of this study suggest that sedation may reduce severe complications compared with general anesthesia in stable conditions. However, the optimal anesthetic method for patients with severe symptoms remains to be determined. Our study also did not compare the incidence of minor complications. Furthermore, the difference in the accuracy of the catheterization results needs to be considered. Positive pressure ventilation during general anesthesia affects hemodynamics through decreased right ventricular preload. Hypercapnia during sedation caused by upper airway obstruction or hypoventilation also affects pulmonary vascular resistance. Therefore, further research on these anesthetic methods is necessary to assess the safety and accuracy of pediatric diagnostic cardiac catheterization in various patients.

Our study has several limitations. First, it was a retrospective observational study based on an administrative claims database. The database did not contain data on several confounding factors, such as medication history, blood coagulation disorder, American Society of Anesthesiologists physical status, number of anesthesiologists in the hospitals, duration of procedure, and operator factors [18]. Although we adjusted for numerous variables, there may be residual confounders. Second, conversion from sedation to general anesthesia could not be identified. The converted patients may have been included in the general anesthesia group because of a rule: a patient can be charged for only one procedure when both general anesthesia and sedation are performed. Assuming an occurrence of conversion of 1% and an occurrence of severe complications in the converted group of 8% (10 times that of the sedation group), approximately 20 patients and two cases of severe complications might be misclassified. However, this potential misclassification would have had a small impact on the results, and the proportion of patients with severe complications would still be higher in the general anesthesia group. Third, we could not exclude outcome events that occurred prior to catheterization on the same day. However, because severely ill patients were excluded, we considered that outcome events were unlikely to occur before the examination. Fourth, this study did not evaluate long-term complications. Finally, the occurrence of different types of severe complications was too infrequent to be compared individually.

In conclusion, this nationwide retrospective study showed that the occurrence of severe complications in pediatric patients (aged < 2 years) who underwent diagnostic cardiac catheterization for VSD with stable conditions was significantly higher in the general anesthesia group than in the sedation group performed under comparable institutional environments. Interventional studies on anesthetic methods for pediatric cardiac catheterization are required to improve the safety and accuracy of the examination.

## Figures and Tables

**Figure 1 jcm-11-05165-f001:**
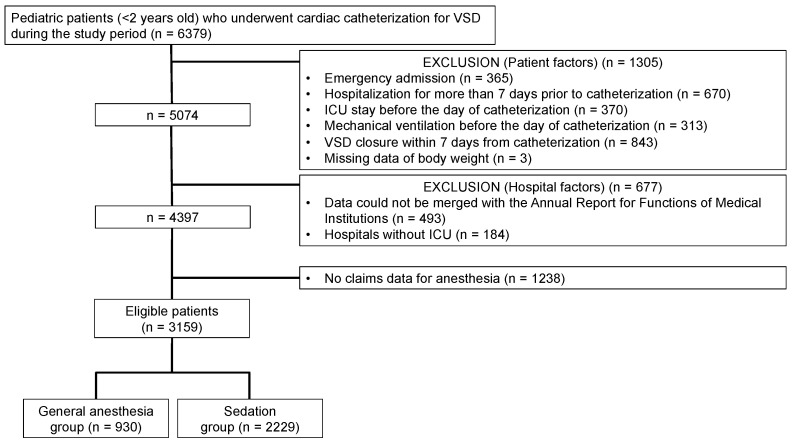
Patient selection flowchart. VSD, ventricular septal defect; ICU, intensive care unit.

**Table 1 jcm-11-05165-t001:** Baseline characteristics of all eligible patients before adjustment.

Variable		General Anesthesia (*n* = 930)	Sedation (*n* = 2229)	Standardized Mean Difference %
Patient characteristics				
Age (months), *n* (%)				
	<3	109 (12)	338 (15)	−10
	3–5	260 (28)	674 (30)	−5
	6–11	561 (60)	1217 (55)	12
Male sex, *n* (%)		441 (47)	1133 (51)	−6.8
Body height (cm), mean (SD)		65 (10.5)	64.5 (11)	4.6
Body weight (kg), mean (SD)		6.5 (2.7)	6.5 (2.7)	0
Comorbidities, *n* (%)				
	Pulmonary hypertension	216 (23)	641 (29)	−13
	Heart failure	319 (34)	977 (44)	−20
	Trisomy (21, 18, 13)	183 (20)	247 (11)	24
Fiscal years of admission *, *n* (%)				
	2010–2011	111 (12)	418 (19)	−19
	2012–2013	182 (20)	548 (25)	−12
	2014–2015	249 (27)	577 (26)	2
	2016–2017	263 (28)	483 (22)	15
	2018	125 (13)	203 (9.1)	14
Hospital characteristics				
Annual number of all surgeries, *n* (%)				
	0–376	446 (48)	978 (44)	8.2
	>376	484 (52)	1251 (56)	−8.2
ICU beds, *n* (%)				
	0–21	257 (28)	1049 (47)	−41
	>21	673 (72)	1180 (53)	41
All beds, *n* (%)				
	0–658	566 (61)	1209 (54)	13
	>658	364 (39)	1020 (46)	−13
Pediatric beds, *n* (%)				
	0	16 (1.7)	218 (9.8)	−35
	1–48	419 (45)	846 (38)	14
	>48	495 (53)	1165 (52)	1.9
Pediatric-related ICU beds, *n* (%)				
	0	162 (17)	185 (8.3)	27
	1–9	300 (32)	1021 (46)	−28
	>9	468 (50)	1023 (46)	8.9
Hospital category, *n* (%)				
	Academic hospital	408 (44)	1146 (51)	−15
	Nonacademic advanced hospital	107 (12)	541 (24)	−34
	Other hospitals	415 (45)	542 (24)	44
Pediatric inpatient management fee ^†^, *n* (%)				
	None	16 (1.7)	218 (9.8)	−35
	1	454 (49)	1059 (48)	2.6
	2	438 (47)	725 (33)	30
	3	15 (1.6)	171 (7.7)	−29
	4	7 (0.8)	56 (2.5)	−14

ICU, intensive care unit; SD, standard deviation. * The Japanese fiscal year begins in April and ends in March. ^†^ Smaller number indicates better-equipped hospitals.

**Table 2 jcm-11-05165-t002:** Baseline characteristics after adjustment by overlap weighting.

Variable		General Anesthesia	Sedation	Standardized Mean Difference %
Patient characteristics				
Age (months), %				
	<3	13	13	0
	3–5	30	30	0
	6–11	57	57	0
Male sex, %		49	49	0
Body height (cm), mean (SD)		64.4 (10.5)	64.4 (11.6)	0
Body weight (kg), mean (SD)		6.4 (3)	6.4 (2.2)	0
Comorbidities, %				
	Pulmonary hypertension	25	25	0
	Heart failure	38	38	0
	Trisomy (21, 18, 13)	18	18	0
Fiscal years of admission *, %				
	2010–2011	13	12	0
	2012–2013	22	22	0
	2014–2015	28	28	0
	2016–2017	26	26	0
	2018	12	12	0
Hospital characteristics				
Annual number of all surgeries, %				
	0–376	43	43	0
	>376	57	57	0
ICU beds, %				
	0–21	37	37	0
	>21	63	63	0
All beds, %				
	0–658	55	55	0
	>658	45	45	0
Pediatric beds, %				
	0	3	3	0
	1–48	42	42	0
	>48	55	55	0
Pediatric-related ICU beds, %				
	0	5.3	5.3	0
	1–9	37	37	0
	>9	58	58	0
Hospital category, %				
	Academic hospital	51	51	0
	Nonacademic advanced hospital	17	17	0
	Other hospitals	32	32	0
Pediatric inpatient management fee ^†^, %				
	None	3.0	3.0	0
	1	58	58	0
	2	35	35	0
	3	2.7	2.7	0
	4	1.1	1.1	0

ICU, intensive care unit; SD, standard deviation. * The Japanese fiscal year begins in April and ends in March. ^†^ Smaller number indicates better-equipped hospitals.

**Table 3 jcm-11-05165-t003:** Comparison of the proportion of patients with severe complications between the general anesthesia and sedation groups, before and after adjustment by overlap weighting using propensity scores.

Analysis	General Anesthesia % (*n*)	Sedation % (*n*)	Risk Difference	95% Confidence Interval	*p*-Value
Before adjustment	2.6% (24/930)	0.8% (17/2229)	1.8%	0.7−2.9%	0.001
After adjustment	2.4% (38/1580)	0.6% (10/1580)	1.8%	0.9−2.6%	<0.001

**Table 4 jcm-11-05165-t004:** Types of severe complications in the general anesthesia and sedation groups, before and after adjustment by overlap weighting using propensity scores.

Type of Severe Complication	Before Adjustment, % (*n*)		After Adjustment, %	
	General Anesthesia	Sedation	General Anesthesia	Sedation
Death	0 (0)	0 (0)	0	0
Surgery for cardiac tamponade	0 (0)	0 (0)	0	0
ECMO management	0 (0)	0 (0)	0	0
Electrical cardioversion	0 (0)	0.09 (2)	0	0.1
Catecholamine use	2.2 (20)	0.6 (13)	1.9	0.4
Pericardiocentesis	0 (0)	0 (0)	0	0
ICU admission	1.9 (18)	0.6 (14)	0.7	0.3
Cardiopulmonary resuscitation	0 (0)	0 (0)	0	0
Composite outcome	2.6 (24)	0.8 (17)	2.4	0.6

ECMO, extracorporeal membrane oxygenation; ICU, intensive care unit.

## Data Availability

There are no supporting data.

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
