# Peer review of "Severe Complications after General Anesthesia versus Sedation during Pediatric Diagnostic Cardiac Catheterization for Ventricular Septal Defect"

_jcm, 2022, doi:10.3390/jcm11175165_

Round 1
Reviewer 1 Report
This paper studied the security problem in general anesthesia versus sedation during pediatric diagnostic cardiac catheterization for the ventricular septal defect. The author should focus more on 1) polishing the article. English language and style are minor spell check required. 2)elaborating that the covariates included. eg: body surface area; Medication history; Vital signs upon admission; Duration of surgery. 3) whether to count the complications of a certain time span after the operation.4) It is proposed to add a chart to illustrate the conclusions.5)The discussion section is not in-depth enough.
Author Response
Thank you for your comments.
1) Our manuscript has undergone another round of English language editing. We will submit the editing certificate for review.
2) We agree that additional covariates may improve the comparison. However, because this study used an administrative database, some variables were not available. We have added information about medication history and duration of procedures in the limitation section (lines 263–265). We did not use body surface area because it is calculated based on height and weight. We designed the study to only include patients with stable conditions. Therefore, we do not believe that vital signs on admission would be a considerable confounding factor.
3) The study evaluated severe complications that occurred within 7 days of catheterization. This is described in lines 121–122.
4) We believe that our results and conclusions are sufficiently summarized in the text, tables, and abstract. Moreover, we are not certain if such a chart would comply with the style of the journal. We would therefore like to ask for the Editor’s recommendation. If an additional chart is required, we will add one.
5) We believe we had written a necessary and sufficient discussion based on the results. We kindly ask the reviewer to indicate the specific parts that require further discussion.
Reviewer 2 Report
I revised Ogawa's manuscript, which compares severe complications following general anesthesia versus sedation, particularly the usage of cathecolamines, during pediatric diagnostic cardiac catheterization for ventricular septal defect. The following are my comments and questions:
1a. The authors define "cathecolamine use" as a periprocedural complication. Were there any long-term adverse effects associated with cathecolamines?
1b. Which catecholamines did you utilise? What was the dosage of cathecolamines? How long was the infusion administered?
2. Please describe the adjustment by overlap weighting in detail:
a. which are the selected covariates ?
b. what is the estimated propensity score for covariates using logistic regression ?
c. how many 1:1 pairs did you find ?
d. did you achieve balance on the covariates through the matching procedure ? (please provide standardized mean differences and the variance ratio or a Kolomgorov-Smirnov test)
Author Response
1a. The authors define “cathecolamine use” as a periprocedural complication. Were there any long-term adverse effects associated with cathecolamines?
Response:We did not have data on long-term outcomes, including adverse effects associated with catecholamines. However, as described below, catecholamines were used for a short period. Therefore, long-term adverse effects may be rare. We have added the following sentence to the limitation section of the revised manuscript: “Fourth, this study did not evaluate the long-term complications” (lines 277–278).
1b. Which catecholamines did you utilise? What was the dosage of cathecolamines? How long was the infusion administered?
Response:Catecholamines included adrenaline, noradrenaline, dopamine, and dobutamine. We have added this information to the Outcomes subsection (lines 124–125).
We did not have a detailed data on the dosage. However, we confirmed the duration of catecholamine use. The median duration of catecholamine use was 3 (interquartile range [IQR], 1.75−4) and 4 (IQR, 2−7) days in the general anesthesia and sedation groups, respectively. We have added these results to the Results section of the revised manuscript (lines 195–197).
2. Please describe the adjustment by overlap weighting in detail:
a. which are the selected covariates?
b. what is the estimated propensity score for covariates using logistic regression?
Response:These are described in the Statistical analysis subsection (lines 130–144). For further clarification, we have added the term “for estimating the propensity scores” (line 134).
c. how many 1:1 pairs did you find ?
Response:We performed a weighted analysis of all patients. Unlike propensity score matching, the number of pairs was not relevant in this study.
d. did you achieve balance on the covariates through the matching procedure ? (please provide standardized mean differences and the variance ratio or a Kolomgorov-Smirnov test)
Response:“Standardized difference” in the original manuscript referred to “standardized mean difference”. We have revised it accordingly. As shown in Table 2, the two groups were balanced well after weighting.
Reviewer 3 Report
This is a well written and (for pediatric cardiology) high volume (n=3159) retrospective analysis of severe complications following diagnostic catheterization for infants (all younger than 12 months) with ventricular septal defect comparing general anesthesia and sedation. The nationwide study is well conducted and provides significant insights that should be published.
The reviewer has nothing to critizise about the study but does not understand why so many infants need diagnostic catheterization in the first place. In the institution of the reviewer and generally in Europe most infants with ventricular septal defect undergo surgery based on solely echocardiographic diagnostics.
Author Response
Thank you for your comments. Preoperative catheterization for VSD is routinely performed in Japan. Although generalizability may be limited, we believe that our findings provide insights into the choice of anesthetic methods for pediatric patients with relatively stable conditions.
Round 2
Reviewer 2 Report
All of the manuscript's arguments are founded on the usage of catecholamines, which was described as a periprocedural complication. Please explain why you regard catecholamine use to be a complication. Please incorporate arguments from other articles into the Discussions section.
Author Response
Thank you for your comment. Complications after pediatric cardiac catheterization were classified by Bergersen et al (reference 2). Use of catecholamines satisfies "Transient change in condition that may be life threatening if not treated, and additional medication was required to return to baseline condition" in the article. We have therefore included it as an outcome. Furthermore, we selected patients with a stable condition undergoing diagnostic cardiac catheterization. These patients usually do not require post-procedural catecholamines.
We analyzed a composite outcome, and the use of catecholamines often occurred concurrently with other outcomes (mainly ICU admission). We believe that the composite outcome illustrates the severe post-procedural condition of patients.
We have revised the Outcomes subsection and Discussion sections referring to previous studies (lines 122-123, and 226-231).